# Parents’ Perception of Staff Support in a Father-Friendly Neonatal Intensive Care Unit

**DOI:** 10.3390/children10040673

**Published:** 2023-03-31

**Authors:** Linn Iren Risanger, Poul-Erik Kofoed, Betty Noergaard, Signe Vahlkvist

**Affiliations:** 1Department of Paediatrics and Adolescent Medicine, Lillebaelt Hospital, University Hospital of Southern Denmark, Sygehusvej 24, 6000 Kolding, Denmark; 2H.C. Andersens Children’s Hospital, Odense University Hospital, 5000 Odense, Denmark; 3Department of Regional Health Research, University of Southern Denmark, 5000 Odense, Denmark

**Keywords:** fathers, mothers, intensive care units, neonatal, infant, newborn, father–child relations, nurse–patient relationship

## Abstract

Healthcare professionals, especially nurses, play a central role in supporting parents during their stay in neonatal intensive care units (NICUs). Fathers often have their own support needs; however, studies have shown that these needs are rarely met to the same degree as those of the mothers. We developed a “father-friendly NICU” with the aim of providing good-quality care to the entire family. To evaluate the impact of this concept, we adopted a quasi-experimental design; using the Nurse Parent Support Tool (NPST), we investigated the differences in the fathers’ (*n* = 497) and mothers’ (n = 562) perception of the nursing support received on admission and discharge between before and after the intervention. In the historical control and intervention groups, the fathers’ median NPST scores at admission were 4.3 (range, 1.9–5.0) and 4.0 (range, 2.5–4.8), respectively (*p* < 0.0001); at discharge, these scores were 4.3 (range, 1.6–5.0) and 4.4 (range, 2.3–5.0), respectively (difference not significant). In the historical control and intervention groups, the mothers median NPST scores at admission were 4.5 (range, 1.9–5.0) and 4.1 (range, 1.0–4.8), respectively (*p* < 0.001); at discharge, these scores were 4.4 (range, 2.7–5.0) and 4.4 (range, 2.6–5), respectively (difference not significant). The parental perception of support did not increase after the intervention; however, the parents reported a high level of staff support both before and after the intervention. Further studies should focus on parental support needs during the different phases of hospitalization (i.e., admission, stabilization, and discharge).

## 1. Introduction

The birth of a sick or preterm infant is often unexpected for the parents; both mothers and fathers commonly experience emotional stress during their stay in the neonatal intensive care unit (NICU) [1,2]. Parents often report being worried about their infants’ condition and chances of survival as well as their own ability to care for their infants. Furthermore, the physical NICU environment is often seen as unfamiliar and distressing; this feeling is reinforced when watching their infant be attached to medical equipment [1,3]. Research has found that forming a parent–child attachment under these conditions can be challenging; this attachment is essential for an infant’s social and intellectual development [4,5,6]. Additionally, the parents are placed at a risk of developing psychological disorders. For instance, it has been shown that the risk of depression is higher in fathers of prematurely born infants than in fathers of infants born at term. Furthermore, the symptoms of fathers who become depressed tend to worsen throughout hospitalization, and even after the infant has been discharged [7]. 

Healthcare professionals, especially nurses, play a central role in supporting the parents during their NICU stay [8,9,10]. However, studies have found that the fathers’ needs are not always met in the NICU. This is attributed to the work culture in the NICU: staff primarily focus on the involvement and support of mothers, unintentionally leaving fathers feeling excluded and vulnerable [11,12]. Previous studies revealed that fathers appreciated being recognized as equal partners in their infant’s care, and wished for information directly from the staff as well as father-specific support. However, they also became confused when the nurses had different approaches to how and when they should involve fathers in childcare or if they perceived the information provided as conflicting [10,11,13,14,15]. When fathers felt that they received satisfactory support, they experienced valuable father–infant bonding and increases in their confidence in the parental role, control, and engagement in childcare; these feelings were maintained even after discharge [16,17]. To meet the wishes and requirements of fathers in NICUs, we have developed and implemented a father-friendly NICU at the University Hospital of Southern Denmark, Kolding [13,18]. More detailed information regarding this father-friendly NICU is provided later in this article.

This paper details a study on the parents’ experience of the concept of a father-friendly NICU, in which the nurses intend to establish a supportive relationship with the parents to help them cope with their infant’s illness and hospitalization. Thus, the aim of this study was to investigate the effects of a father-friendly NICU on the parents’ perceptions of staff support.

## 2. Materials and Methods

### 2.1. Study Design and Setting

In this quasi-experimental study, we investigated the parents’ perception of staff support in a father-friendly, 22-bed NICU. This study was conducted at the Department of Pediatrics and Adolescent Medicine, University Hospital of Southern Denmark, Kolding. The neonatal unit was a level II/III NICU [19] that provided care to infants born at ≥28 week of gestation (independent of the weight) and short-term mechanical ventilation support until stabilization for transfer. Approximately 600 infants are admitted to this unit per year, mainly from the delivery room; however, some are also admitted either from the maternity ward or from home.

Before the intervention, the infant’s parents and siblings had unlimited access to the NICU. Armchairs were placed next to the incubators and cradles. The parents could board an adjacent hotel for sleep at night if they wished. One parent could stay free of charge. However, if the infant was hospitalized for a longer period, both parents could stay with their infant in the family room without having to pay.

### 2.2. Sample

Infants admitted to the NICU between December 2011 and December 2012, and their parents, were recruited as the historical control group; infants admitted between August 2013 and May 2015, and their parents, were recruited as the intervention group (Figure 1). The exclusion criteria were as follows: (a) neither parent understood spoken nor written Danish, (b) the infant was critically ill (such infants were transferred to a level III NICU), (c) the mother was critically ill, (d) the infant was admitted from home, and (e) the infant had a single parent. In case of twins, the parents were included just once and data were recorded only for the more severely ill twin.

### 2.3. Intervention

A participatory action research was conducted from August 2011 to July 2013 to develop the “father-friendly NICU”; it involved the participation of fathers, mothers, interdisciplinary health professionals, and managers. The development process has been described in detail elsewhere [18] (ClinicalTrials.gov; NCT05521620). In this research, qualitative methods were used to obtain a broader understanding of the fathers’ needs when their infants were admitted to the NICU. The key finding was that the fathers were stressed mainly because they were worried about their newborns and partners. Furthermore, they were torn between being in the hospital with the newborn and mother, going to work, and taking care of their older children. They balanced between being breadwinners and coparents on an equal footing with the mothers. The fathers wanted to be the stronger of the two parents, so that they could support their partners. They felt ignored because they thought that the staff mainly focused on the mothers and the infants. Therefore, they wished to receive relevant information directly from the staff, rather than from the mothers. Furthermore, they wished to have an outlet for freely expressing their feelings and concerns.

Based on the findings from the participatory action research, eight core principles of a father-friendly NICU were developed and implemented (Box 1). Fathers wanted to be equal co-parents and wanted to participate in the important events of their infants’ lives (such as the first bath or the transfer from an incubator to a cradle). However, these events were often scheduled during the daytime, when the fathers could not be present. Therefore, we suggested that the staff should plan important events in which fathers could participate, if possible. Furthermore, when present in the NICU, the fathers were encouraged to engage in skin-to-skin contact with their infants. 

Box 1The 8 core concepts of a father-friendly NICU.
Staff shall encourage fathers to have skin-to-skin contact with their infants as soon as the infants are admitted to the NICU if the mothers are still in the recovery room or intensive care unit.Fathers shall be encouraged and given the opportunity to participate in important situations for their infants, such as the first bath, being moved from the incubator to the cradle, etcetera.Fathers shall receive information and guidance directly from the healthcare professionals (and not only through the mothers).Medical rounds with neonatologist regarding important information about the infants’ condition and development shall be scheduled so both parents can participate.The department shall organize mother and father groups where the parents can talk parent-to-parent about their situation.The families shall have the opportunity to have a close family member to support them in the unit. The family member can stay with the infant if the parents want, e.g., if the parents want to visit their older children at home.Older siblings shall have the opportunity to stay overnight.The department shall offer counseling by a social worker about paternity leave and other social/economic issues.


The fathers stated that they invested considerable energy into finding information about paternity leaves. Furthermore, they expressed feeling that they had not received sufficient assistance and guidance from the nurses, leading them to vacillate among several responsibilities, namely being at the hospital with the mother and the infant, staying at home to look after the older children, and attending work. To address these dilemmas, we allowed older children to stay overnight at the hospital in a hotel room with their parents. Furthermore, a social worker was made available in the NICU for counselling the fathers on the possibility of taking a paternity leave and other social and economic issues. When the father had to leave, a close family member was allowed to stay with the mother and the infant at the hospital.

The fathers indicated that they greatly needed to discuss their concerns. However, they also admitted that it was challenging to talk about their own feelings, because they wanted to spare the infant’s mother the worries. Based on the suggestions obtained during the participatory action research, the NICU staff created groups for fathers facilitated by experienced nurses. 

Finally, the fathers’ need for information was addressed. Generally, if the fathers were absent from the NICU during medical rounds, they received all information about their infants’ condition and health from the mothers. However, the fathers wished to be informed by the staff caring for the infants and the mothers. Therefore, it was decided that nurses should directly guide and inform the fathers whenever they are present in the NICU. Furthermore, neonatologists should plan medical rounds that allow both parents to participate. 

To ensure that the staff understood and accepted the core principles of the father-friendly NICU, different activities were conducted for their engagement and education. Traditionally, the staff have focused on mothers and infants; however, they are now required to incorporate a more father-friendly approach into their routines. Because nurses played a key role in the implementation of this father-friendly NICU, they were educated on the traditional and cultural norms of fathers and men and on the dilemma between being breadwinners and equal co-parents. A project nurse carried out bedside practice with all the nurses. At daily nursing conferences, reflections were conducted to support the staff in improving their collaboration with the parents. Furthermore, the researcher (second author) was available in the NICU on every weekday and was in an ongoing dialogue with the staff about the fathers’ needs and requests and whether these were met. Additionally, focus groups were established with the aim of reflecting on the participants’ experiences before and after the implementation of the father-friendly NICU (manuscript under review).

None of the researchers participated in the care or treatment of the families in the historical control and intervention groups.

### 2.4. Outcomes

The primary outcome was the difference in the parents’ perception of the staff support received on admission and discharge between the historical control and intervention groups. The secondary outcomes were the predictors of the support received. 

### 2.5. Instruments

A Danish version of The Nurse Parent Support Tool (NPST) [20] was used to evaluate the parents’ perception of the nursing support received during their infants’ hospitalization (Appendix A). After obtaining approval from Miles et al. [20], the NPST was translated from English to Danish, and then back-translated to English in accordance with the ISPOR recommendation [21]. The questionnaire was tested qualitatively for content validity by 9 fathers and 12 mothers and adjusted for minor linguistic changes in accordance with their answers. 

The NPST includes the following four dimensions of nurse support: (1) communication of information related to the infant’s condition and care (nine items), (2) support mainly directed at enhancing the parental role (four items), (3) emotional support to help parents cope with their infant’s sickness (three items), and (4) caregiving support concerning the quality of care provided to the infant (five items). For each item, the parents scored the degree of support received on a 5-point Likert scale, with scores ranging from 1 (almost never) to 5 (always); the higher the score, the greater was the perceived support from the nursing staff [20]. 

### 2.6. Data Collection

The participating parents received up to four questionnaires depending on the length of their infant’s NICU stay. We decided to set a minimum interval of 5 days between follow-up questionnaires to avoid parental questionnaire fatigue. Thus, parents received a questionnaire on the day of admission (+2 days), on day 14 (±3 days), at discharge (for infants hospitalized for more than 5 days), and at one month after discharge.

The first questionnaire was administered to the parents in an envelope with an information sheet. They could choose to complete subsequent questionnaires, which were either paper based or electronic. Electronic questionnaires were administered using SurveyXact (http://www.surveyxact.com/, accessed on 10 January 2020); paper-based questionnaires were also entered in SurveyXact. Members of the historical control group were enrolled during the period when the concept of the father-friendly NICU was developed (Figure 1). All data from the historical control and intervention groups were analyzed at the end of the study period.

### 2.7. Statistical Analysis

Data from the parents who completed at least one questionnaire were included in this study. In the admission questionnaire (Q1), the parents rated staff support received during the first few days of their infant’s hospitalization. Data from Q1 and the last questionnaire completed during hospitalization, either on day 14 (Q2) or at discharge (Q3), were analyzed. For group comparisons, a two-sided Student’s *t*-test and a Wilcoxon’s rank-sum test were used for normally and non-normally distributed data, respectively. Chi-square tests were used to assess binary outcomes. Summary statistics are presented as mean (standard deviation (SD)) for normally distributed data and as median (range) for other data.

Predictors of NPST results were analyzed using a multiple regression model, with the NPST as the dependent variable and gestational age, cesarean section, neonatal pulse oximetry, oxygen therapy, intravenous line, parental age, children at home, parental occupation, and the study group as the independent variables. The residuals of the models were checked for a normal distribution using q-norm plots. The model was run separately for mothers and fathers for the Q1 and Q2/Q3 answers. For all statistical analyses, the significance level was set at *p* < 0.05. Statistical analyses were performed using STATA version 16 (StataCorp, College Station, TX, USA).

### 2.8. Ethics

According to Danish legislation, this study did not require permission from the Regional Committees on Health Research Ethics. The Danish Data Protection Agency approved this study. Parents received oral and written information about the study before participation. Participation was voluntary, and the participants could withdraw at any time without any consequences on the quality of care of their infants. All procedures were performed in accordance with the principles of the Helsinki Declaration.

## 3. Results

### 3.1. Participants

Data from 497 fathers and 562 mothers were analyzed. The historical control group comprised 234 fathers and 252 mothers, while the intervention group comprised 263 fathers and 310 mothers. The demographic data of both the parents and their infants are provided in Table 1. Significantly more male infants were born during the intervention period (*p* = 0.05). Furthermore, there were no statistical differences in the descriptive data between the historical control and intervention groups.

Legend: demographic data of fathers, mothers and infants in the intervention group and the historical control group. In the intervention group perisignificantly more infants were boys. No other differences was observed between the groups.

### 3.2. Fathers’ Perceived Support Score

The admission questionnaire (Q1) was completed by 218 and 229 fathers in the historical control and intervention groups, respectively. The discharge questionnaires (Q2/Q3) were completed by 87 and 112 fathers in the historical control and intervention groups, respectively. Furthermore, the median NPST scores at admission were 4.3 (range, 1.9–5.0) and 4.0 (range, 2.5–4.8) in the historical control and intervention groups, respectively (*p* < 0.0001); at discharge, these were 4.3 (range, 1.6–5.0) and 4.4 (range, 2.3–5.0), respectively; this difference was non-significant (Figure 2). Among the variables included in the multiple regression model, pulse oximetry-based monitoring of the infant increased the fathers’ NPST scores at admission (*p* = 0.045) but not at discharge. Other factors influencing the fathers’ NPST scores at admission were age (*p* = 0.04), number of children at home (*p* = 0.03), and length of NICU stay (*p* = 0.000).

### 3.3. Mothers’ Perceived Support

The admission questionnaire (Q1) was completed by 232 and 278 mothers in the historical control and intervention groups, respectively. The discharge questionnaires (Q2/Q3) were completed by 116 and 147 mothers in the historical control and intervention groups, respectively. The median NPST scores at admission were 4.5 (range, 1.9–5.0) and 4.1 (range, 1.0–4.8) in the historical control and intervention groups, respectively (*p* < 0.001). At discharge, these were 4.4 (range, 2.7–5.0) and 4.4 (range, 2.6–5.0), respectively; this difference was not significant (Figure 2).

Of the variables included in the multiple regression model, only pulse oximetry-based monitoring of the infant increased the mother’s NPST score at admission (*p* = 0.007) but not at discharge.

## 4. Discussion

We developed a father-friendly approach to parental support in the NICU. It comprised different initiatives to address the needs of fathers during the hospitalization of their infants [18]. This study explored the impact of this parental care model on the fathers’ and mothers’ perceptions of the support received from the NICU staff.

Before the intervention, the parents experienced a high degree of nursing support in the NICU. The mean total NPST scores for the fathers and mothers at admission were 4.3 and 4.5, respectively. These scores indicate the parents’ perception of having received nursing support, either most of the time or always, across all four NPST support dimensions (informational, emotional, parental role, and caregiving support).

Surprisingly, after the approach was implemented, both fathers and mothers had a significantly lower perception of nursing support at admission. This finding might be explained by the results of a previous qualitative review and meta-synthesis of the parents’ and nurses’ experiences in NICUs; this review revealed that parents could not effectively collaborate with the staff during the initial days of hospitalization [9]. During this phase, the parents were worried, felt out of control of the situation, and preferred to be passive observers. An essential factor for strengthening the nurse–parent relationship at admission is to keep the parents informed of their infant´s condition and encourage skin-to-skin contact between the two [4]. After our father-friendly approach was implemented, the staff may have attempted to get the fathers more involved in their infants’ care early in the hospitalization period with good intentions in mind. However, the fathers could have perceived this as receiving lesser support; if some fathers were not mentally capable of handling active involvement at this time, they could have felt a lower satisfaction with the staff support received.

Our finding of a lower perceived support from the nurses does not correspond with the findings of a study by LeDuff et al. [16] which revealed a nonsignificant increase in the perceived nursing support after an educational intervention was undertaken to improve paternal support. The fact that the fathers in our study reported a lower nursing support after the intervention might be further explained by the results of our previous study on paternal stress in the NICU [22]: in parallel with the present study, we also investigated the feeling of stress in 109 fathers in the NICU before and after a father-friendly intervention. These fathers had higher stress levels because they were more involved in the caring of their infants at both admission and discharge. The fathers’ involvement, staff’s expectations, and social expectations to fulfil the traditional breadwinning and caregiving roles may have caused this increased stress. Increase in stress with greater involvement may elicit the need for greater patient education on stress-management strategies, as studies have found that psychotherapy-based interventions (such as mindfulness, lessons in relaxation and self-knowledge, and father-support groups) decrease paternal symptoms of anxiety and stress [7]. 

Upon discharge, there was a catch-up in the total NPST scores in the post-intervention group for both mothers and fathers. This might be explained by a lower parental stress level later during hospitalization, and thus, a better parental ability to cope with the expectation of involvement [9,22]. The perceived nurse support at the time of discharge, from the point of view of both the fathers and the mothers, was the same between before and after intervention. Therefore, the nurses’ emphasis on getting the fathers involved in infant care initially may have led the parents to perceive that they were receiving lesser support. However, following admission, the parents grew more confident about the fathers’ role and appreciated the nurses’ help.

Among the fathers, the number of children at home was negatively correlated with the perceived staff support. This finding is in concordance with the findings of an integrated analysis of paternal needs in NICUs, which revealed that fathers often struggle to balance their responsibilities at home and work with the time spent in the NICU [23]. Most fathers desire to obtain information directly from medical staff and wish to attend medical rounds; however, this is not always possible, because fathers are also required to work and care for the other children at home [8,11]. A solution to this may be an increased use of technical aids (such as video-based patient rounds) to enhance the fathers’ participation in staff communication [11]. 

In our study, the perception of staff support was higher in older fathers than in younger fathers. We hypothesized that this was partly due to the fact that older fathers had a greater life experience and ability to cope with stressful events; a recent study found that younger fathers felt more stressed during their stay in NICUs [24]. 

Interestingly, parents of infants who were not monitored using pulse oximetry at admission reported a significantly lower level of staff support. One might assume that this subgroup of neonates was less sick or more mature than their counterparts; thus, the staff may not have payed a close attention to these infants or their parents. However, research suggests that NICU admission is a great source of emotional strain for the parents [1,3]. The change in the parental role associated with NICU hospitalization was the leading cause of stress for both parents, and parental stress was partially independent of the neonates’ characteristics and clinical conditions. It is recommended that parental stress should not be underestimated in the presence of a low neonatal risk and that support for parents should be organized and delivered independently of the infant’s clinical condition [1]. 

### Strength and Limitations

Our study has some strengths. First, we implemented a multifaceted intervention program, which was developed in partnership with the fathers, mothers, and staff members to ensure that the changes were tailored to the NICU. Second, the sample size of our study was large, and the historical control group served as a baseline comparator in our quasi-experimental study. Third, the intervention was implemented and evaluated in real-life clinical practice. 

However, this study also has some limitations. First, this was not a randomized controlled trial, since participants were selected for either the historical control group or the intervention group based on their hospitalization period. However, it would not have been possible to adopt a father-friendly approach for the parents of one infant while simultaneously avoiding the same for the parents of another infant in the same unit. Second, due to the study design, the staff were not blinded to the intervention because this was not practically achievable. Third, there was an overlap between the parents’ enrolment in the historical control group and the development of the principles of the father-friendly concept. The fact that the nurses and other staff members were involved in the process of improving the fathers’ condition in the unit may have biased the parents’ perceptions of nurse support in a positive direction. Fourth, the nurses’ attitudes towards the fathers may have been influenced to turn more father-friendly in the historical control group even before the intervention was implemented in the unit. However, our nonrandomized study was strengthened by explicit inclusion and exclusion criteria. The study lasted for over three years, and changes other than the intervention could have influenced the parents’ perceived support [25]. Nevertheless, there were no other changes in the importance of the NICU during that period.

Fifth, by assessing only the perception of support, we may have missed other effects on the paternal mental health. Several other aspects, such as well-being, attachment, empowerment, and self-efficacy, were not assessed.

Finally, our sample did not include parents of extremely preterm infants (gestational age < 28 weeks). The support needs of such fathers may be different from those of the fathers of more mature infants. 

## 5. Conclusions

Before the implementation of a father-friendly approach in the NICU, fathers and mothers perceived a high staff support during admission and discharge. However, after implementation, the nurses’ support was scored to be low at admission; this may be explained by the pressure felt by the fathers to be more involved in the care of their infants, thereby creating higher parental stress levels. However, at discharge, the perceived staff support increased and was equal to that in the pre-intervention group, probably because the fathers got used to their new roles. 

Parents of children who were not monitored on admission experienced a lower degree of staff support, which could indicate that the nurses were more focused on sicker infants. Likewise, fathers having more children to care for at home experienced lesser support from the staff, probably due to conflicts between being involved with the newborn infant and taking care of the older children at home.

The findings of this study indicate that in clinical practice, emphasis should be placed on the following: (1) balancing staff support and expectations of paternal involvement with the parents’ emotional stress levels and (2) acknowledging the support needs of parents of less sick infants and fathers with more children at home. There is a need for ongoing research on how parents of infants admitted to the NICU are supported. The present study contributes to the knowledge of interventions that support parents, and our partly negative results highlight the complexity of providing ideal support to parents during a period in which both emotional and informational needs often evolve continuously. More research is needed to elucidate how the parental need for staff support changes throughout the different phases of their infant’s hospitalization and whether individualized stress-management tools may improve the parents’ perception of the support received during their stay in the NICU.

## Figures and Tables

**Figure 1 children-10-00673-f001:**
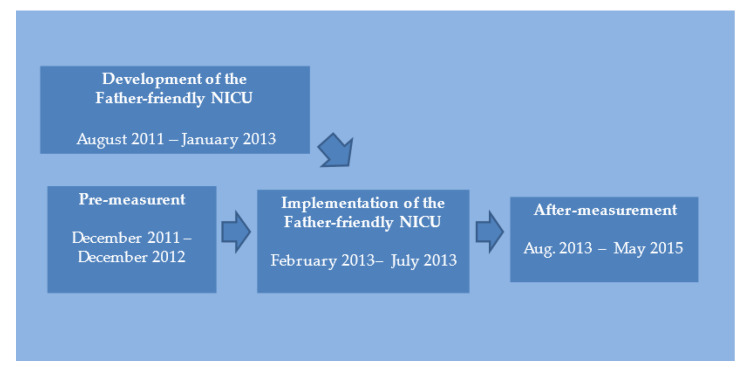
The timeframe of developing the father-friendly NICU.

**Figure 2 children-10-00673-f002:**
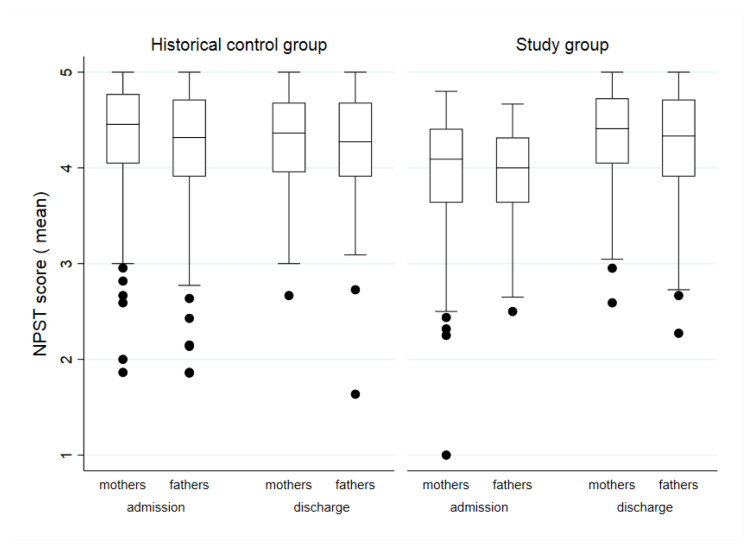
Boxplot of mean NPST score for mothers and fathers before and after the father-friendly intervention.

**Table 1 children-10-00673-t001:** Demographic and clinical data on the participants.

Fathers	Historical Control Group n = 234	Intervention Group n = 263
Age, years (mean, SD)	32.4 (5.2)	32.3 (5.4)
No. of children (mean, SD)	1.5 (0.5)	1.4 (0.5)
No. of other children living with the family (mean, SD)	1.3 (1.1)	1.1 (1.1)
Previously had an infant admitted to a NICU (n, %)	27 (14%)	29 (14%)
Cohabiting with the infant´s mother (n, %)	209 (98%)	222 (97%)
Employment (n, %)		
1. Under education	11 (5%)	13 (6%)
2. Employed	168 (79%)	195 (85%)
3. Unemployed	10 (5%)	3 (1%)
4. Parental leave	12 (6%)	11 (5%)
5. Other	11 (5%)	7 (3%)
**Mothers**	n = 252	n = 310
Age, years (mean, SD)	30.0 (4.8)	30.0 (5.0)
No of children (mean, SD)	1.5 (0.5)	1.4 (0.5)
No. of other children living with the family (men, SD)	1.3 (1.0)	1.3 (1.2)
Previously had an infant admitted to a NICU (n, %#)	32 (15%)	36 (14%)
Cohabiting with infants’ father (n, %#)	220 (95%)	272 (98%)
Employment (n, %#)		
1. Under education	12 (5%)	24 (9%)
2. Employed	43 (19%)	67 (24%)
3. Unemployed	21 (9%)	22 (8%)
4. Parental leave	149 (65%)	152 (86%)
5. Other	6 (3%)	12 (7%)
**Children**	n = 263	n = 319
Boys (n, %)	171 (65%)	181 (57%) *
Twins (n, %)	27 (10.6%)	25 (8.0%)
Gestational age, weeks (median, range)	37 (27–42)	38 (25–48)
Birth weight, kg (median, range)	2.90 (0.91–5.12)	2.92 (0.59–4.98)
Length at birth, cm (median, range)	50.0 (34.0–57.0)	50.0 (30.0–62.0)
Head circumference at birth, cm (median, range)	34.0 (21.0–49.0)	34.0 (22.5–39.0)
Caesarean section delivery (n, %)	118 (45%)	137 (43%)
Apgar score at 5 min (mean, SD)	9.0 (1.6)	9.3 (1.5)
Intervention on the first day of admission (n, %):		
Oxygen therapy	111 (43%)	118 (38%)
Sat monitoring	243 (93%)	290 (92%)
Intravenous access	93 (36%)	95 (30%)
Length of NICU stay, days (median, range)	7 (0–88)	7 (0–125)

# Percentage of answers. * *p* = 0.05.

## Data Availability

Not applicable.

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
