# Peer review of "Parents’ Perception of Staff Support in a Father-Friendly Neonatal Intensive Care Unit"

_children, 2023, doi:10.3390/children10040673_

Round 1

Reviewer 1 Report

1.       What is the main question addressed by the research?:

Father’s perception of nurse support  in the neonatal unit: results of a before and after intervention trial

2.       Do you consider the topic original or relevant in the field? Does it address a specific gap in the field?

Yes, the subject is important, fathers are half of the parents! By assessing only the perception of support, the authors might have missed other effects on paternal mental health.

3.       What does it add to the subject area compared with other published material?

The results of this negative trial open the road to other interventions, important in this field. Interventions targeting parents may have long lasting impact on the development of preterm children for example.

4.       What specific improvements should the authors consider regarding the methodology? What further controls should be considered?

The authors could have assessed other domains, such as well being, attachment, empowerment, self-efficacy etc.

Length of stay was associated with the outcome, how were the results in the preterm population?

5.        Are the conclusions consistent with the evidence and arguments presented and do they address the main question posed?

Yes. This negative study stopped in 2015, what have you implemented since?

6.       Are the references appropriate?

Yes.

You could add this reference on fathers in the NICU:

Psychosocial interventions and support programs for fathers of NICU infants - A comprehensive review, Early Hum Dev, . 2021 Mar;154:105280. doi: 10.1016/j.earlhumdev.2020.105280. Epub 2020 Nov 16.

Line 171: length of admission should be length of stay

Figure 1: is maj may?, this figure should be edited.

Table 1: were the variables for the infants normally distributed? For gest age and measurements, could you add the range, or give median Q1Q3?.

Reviewer 2 Report

Abstract:

It is not clear what the design of the study is.

1. Introduction

A study found that fathers’ needs were not always met in the NICU”. Cite this study/integrate it better within the next sentence.

Paragraph about your qualitative study to develop the intervention could be omitted or just provided in the methods section. State previously described and cite if including here.

2. Materials and Methods

Confusing about the term ‘control group’ – infers that these were observed at the same time as the intervention group. They should be called the pre-intervention group or use the term ‘historical control’. For pre-post studies, a variable of interest is measured before and after an intervention in the same participants – you had different participants. Consider if this is quasi-experimental design.

Control group – assuming they were provided with “standard” care? What is standard care prior to the intervention?

Intervention - some of the 8 core concepts are vague/non-specific., e.g., “meaningful conversations will be scheduled”. Also, was a social worker employed prior to the intervention? This is one core concept of a father-friendly NICU that is part of standard care. How did you ensure all nurses/clinicians provided a father friendly NICU? Was there specific training and ongoing surveillance?

If you are comparing 2 groups, you need to consider any statistical difference in demographic and clinical data by multivariate regression techniques to adjust for differences in baseline characteristics. Consider potential confounders.

Sample – how were the participants allocated i.e. randomization? This is explained in the limitations only. How was the sample size calculated?

Spelling errors in Figure 1.

With the NPST tool, explain that the higher the number, the greater the perceived support.

What Post hoc residual diagnostics were used e.g. Fisher’s, ANOVA?

Identify what determines statistical significance.

Why wasn’t breast feeding included as a variable given it improves infant bonding and parents may need additional support?

If admission scores were analysed before post-intervention scores, how was any bias controlled? For example, were the researchers also providing patient care? Consider potential bias.

3. Results

Table 3 has not been provided as referred to in the results section. More results are required other than the mean and SD. Provide the post hoc residual diagnostics, you have just stated not significant.

Figure 2 – the y axis requires more information i.e. NPST (mean), or provide a key for the figure. Detail what type of figure this is and an explanation underneath the figure.

4. Discussion

Interesting that father’s perceived support was statistically lower in the intervention group. You have referred to your own study here but explore other literature around the phenomenon of father’s involvement in NICU care. 

You need to critically analyse the current body of knowledge on this topic and make recommendations for further research and practice implications.

Academic writing: Appreciate translation/English as a second language here but requires improvement so that the academic “voice” is more formal.

Reviewer 3 Report

Dear Authors.

I would like to congratulate on your research work titled Parents’ perception of staff support in a father-friendly neona-tal intensive care unit.

However, some minor revisions to the article must be done.

1.  Introduction:  the background of the previusly published papers in this field is too concise. Please supplement this part of the manuscript.

Study aim is to concise, too. Please, reformulate.

2. M&M. Did authors of a Danish version of The Nurse Parent Support Tool perform any other analysis apart from the content validity? e. g. internal consistency, test-retest reliability, floor and ceiling effect?

Please elaborate more on the statistical methods.

Results. I feel the analysis regarding associations between selected clinical and sociodemographic data, and the results of questionnaires, is missing. 

Discussion. Could the authors elaborate more about future research implications?

Round 2

Reviewer 2 Report

This manuscript could be further strengthened with academic writing support, e.g. Page 2, line 41 “getting psychological disorders" – consider the term “developing”; “boys” versus males (line 221).

Lines 90-95: Is this information related to standard care or the intervention?

Provide a succinct overview of the key findings from the action research project that informed the intervention, this consumes most of the information provided on the intervention.

More detail is required regarding the intervention. The key concepts remain vague despite the authors attempting to provide furhter clarification. E.g. “Different activities were conducted to engage and educate the nurses e.g., bedside practice and daily reflection at nursing conferences were conducted supporting the staff to better collaborate with the parents.” The intervention needs to be repeatable for transparency/rigour in research and based on the little detail provided, I cannot see how this could be in its present format.

I still cannot see 'table 3' in the manuscript, which you refer to in-text.

Round 3

Reviewer 2 Report

Abstract

Lines 10-11 – parents don’t “stay in a NICU’.. their baby does.

11 – Fathers often have their own support needs in comparison to mothers.

Introduction

31: Remove ‘the’ unexpected for parents.

34: care for their infant (single).

37: parent-infant attachment

44 – their infant’s NICU stay

52 – Use another term for childcare. What do you mean by this, “or if they perceived the information provided as conflicting”. Conflicting to what the mothers receive?

55 – Use another term for childcare

56 – "needs" instead of wishes and requirements

56 – we developed.

58 – Remove more info regarding this is provided later. Reference the paper after Kolding.

Materials and Methods

70: The neonatal unit is..

74: are they admitted from other hospitals?

76: Use cot instead of cradle.

78: At what period of days admitted can the parents stay for free?

87: In the case of twins…

99: Use another phrase for “equal footing”

108: Cradle

112: infant

Box 1. Concept 2. Please remove the term etcetera

Box 1. Concept 4. Neonatologists

Box 1. Concept 5. Talk to other parents

115. Paternity leave.

116 Another term for vacillate

121 Taking paternity leave

124-128: Instead of repeating the findings, explain what social groups were established. For example, a weekly father’s group in the NICU.

129-134: And again here. If fathers are at work during the day, were medical rounds conducted at night? You have said that they “should plan” medical rounds that allow parents to participate.

147-149: This seems out of place here.

150-151: This sentence needs to be integrated into a paragraph (avoid 1 sentence paragraphs).

191: Normal and non-normal distribution

206: verbal not oral

207: parents were advised that they could withdraw

Results

213: Comprised of..

215: more male infants were admitted

222: change perisignificantly – I’m not aware of this word

Discussion

266: attempted to foster/promote father involvement

268-269: Consider the term “hesitant”. They may have been hesitant to be engage actively in their infant’s care.

271: Conflict instead of do not correspond

291: promoting/encouraging instead of getting them to..

296: In accordance

303: Or non-traditional medical rounds such as at night-time?

306. New sentence; A recent study found that..

311 Close attention.
